# Enterococci Isolated from One-Day-Old Chickens and Their Phenotypic Susceptibility to Antimicrobials in the Czech Republic

**DOI:** 10.3390/antibiotics12101487

**Published:** 2023-09-27

**Authors:** Jaroslav Bzdil, Vladimir Sladecek, David Senk, Petr Stolar, Zuzana Waicova, Nela Kollertova, Monika Zouharova, Katarina Matiaskova, Petr Linhart, Katerina Nedbalcova

**Affiliations:** 1Ptacy S.R.O., Valasska Bystrice 194, 756 27 Valašská Bystřice, Czech Republic; vetmed@seznam.cz (J.B.); sladecek.vladimir@gmail.com (V.S.); dsenk@post.cz (D.S.); ptacy@ptacy.cz (P.S.); 2Department of Cell Biology and Genetics, Faculty of Science, Palacký University Olomouc, 17. Listopadu 1192, 779 00 Olomouc, Czech Republic; zuzka.w28@gmail.com (Z.W.); nela.kollertova02@upol.cz (N.K.); 3Veterinary Research Institute, Hudcova 296/70, 621 00 Brno, Czech Republic; monika.zouharova@vri.cz (M.Z.); katarina.matiaskova@vri.cz (K.M.); 4Institute of Animal Protection and Welfare and Public Veterinary Medicine, University of Veterinary Sciences, Palackeho 1–3, 612 42 Brno, Czech Republic; linhartp@vfu.cz

**Keywords:** animal, poultry, organ, spectrum, prevalence, resistance, multiresistance

## Abstract

Our study describes the prevalence and spectrum of enterococci isolated from one-day-old chickens in the Czech Republic, their level of antimicrobial resistance, and the occurrence of multiresistance. Over a 24-month period from 1 August 2021 to 31 July 2023, a total of 464 mixed samples of one-day-old chicken organs were examined during routine inspections at 12 randomly selected poultry farms in the Czech Republic. The samples were processed via cultivation methods and suspected strains were confirmed using the MALDI–TOF Mass Spectrometry method. Antimicrobial susceptibility was determined using the MIC method for eight antimicrobials. A total of 128 isolates (prevalence of 27.6%) representing 4 species of enterococci were isolated, including *Enterococcus faecalis*, *Enterococcus faecium*, *Enterococcus gallinarum,* and *Enterococcus hirae,* with prevalence rates of 23.3%, 1.5%, 2.2%, and 0.6%, respectively. Susceptibility tests showed a high percentage of susceptible strains among *E. faecalis*, *E. faecium*, and *E. gallinarum* for penicillin-based antibiotics, sulfamethoxazole with trimethoprim, and florfenicol (80–100% susceptible strains). *E. hirae* was an exception, displaying complete resistance to enrofloxacin (0% susceptible strains) and a high degree of resistance to other tested antimicrobials (33.3% susceptible strains). Among the isolated strains, a total of 16 isolates (12.5%) showed resistance to 3 or more antimicrobials. Complete resistance to all eight antimicrobials simultaneously was observed in four isolates (3.1%). This research shows the possible sources of pathogenic enterococci and their virulence and resistance genes. The findings hold relevance for both veterinary and human medicine, contributing to a better understanding of enterococcal circulation in the human ecosystem and food chain, as well as the development of their resistance and multiresistance.

## 1. Introduction

Enterococci are ubiquitous Gram-positive, facultatively anaerobic bacteria that are part of the commensal microflora in the gastrointestinal tract of humans and animals [1,2]. These microorganisms have been isolated from soil, surface water, and seawater and they are also associated with plants and fermented food products and are part of the intestinal microflora of vertebrates and invertebrates [3]. Representatives of the genus *Enterococcus*, which currently includes about 40 recognized species, were originally classified as group D streptococci. Currently, *Enterococcus* spp. are considered a faecal indicator for monitoring microbial sources of contamination [4,5,6]. They are also often used to monitor trends in antimicrobial resistance [7].

Positive attributes of enterococci include the production of various useful substances applicable to food biotechnology, particularly the production of fermented milk products. They tend to be producers of organic acids, ethanol, hydrogen peroxide, carbon dioxide, diacetyl, and most notably, bacteriocins [8,9]. Among these bacteriocins, enterococci generate enterocins A, B, and P, which exhibit an antimicrobial effect against pathogens such as *Listeria monocytogenes*, *Staphylococcus aureus*, *Clostridium* spp., and mundticin KS produced by both *E. mundtii* and *E. faecium* [10].

On the other hand, enterococci can act as pathogens in humans and animals due to the production of certain virulence factors, such as haemolysin, gelatinase, aggregation substance, hyaluronidase, microbial surface components recognizing adhesive matrix molecule (MSCRAMM), cell wall carbohydrates, and capsular polysaccharides [11]. Some strains of enterococci are capable of forming biofilms [12,13]. Over the past two decades, enterococci have been recognized as important human pathogens [14,15,16,17]. They have been implicated in nearly 12% of all nosocomial infections, with approximately 90% of these infections caused by only two species, *E. faecalis* and *E. faecium* [6,18,19]. These two species have been found to be the third and fourth most prevalent conditional human pathogens worldwide [20] and together they rank third among the identified causative agents of bacteraemia in humans in Europe and North America [21,22].

In poultry, enterococci are implicated in the deaths of one-day-old chickens, and in adult poultry, they can cause bacteraemia, pulmonary hypertension, amyloid arthropathy, encephalomalacia, and other neurological disorders [23,24]. Enterococcal infections can be caused by embryo and young bird contamination, following faecal contamination of hatching eggs. However, most transmission occurs through aerosols, ingestion, or skin lesions. One of the greatest risks to egg and chicken quality is bacterial contamination of hatching eggs, which can result in reduced hatchability and increased mortality in the first week of life [25]. Poor chicken quality and a problematic start on the farm can negatively impact flock production [26]. For over 20 years, it has been known that enterococci can acquire resistance to antibiotics relatively easily and can transfer resistance genes to other microorganisms [16]. Enterococci resistant to multiple antimicrobials, including vancomycin, are commonly isolated from humans, wastewater, aquatic habitats, agricultural waste, and animal sources, suggesting their potential to enter the human food chain [17,18]. Various scientific studies report different levels of enterococci resistance to antimicrobials. In the case of poultry, resistance to penicillin in isolates from chicken meat products in the USA has been reported to range from 14.1% to 87.5%, while resistance to amoxicillin with clavulanic acid ranges from 0% to 37.5% [19]. The use of antimicrobials in chickens selects for resistant bacterial populations in the gut microbiota of chickens, which are subsequently spread to the environment via faecal matter [27]. In addition, resistant bacteria carrying resistance genes are able to persist in the poultry gut and environment for long periods of time after antimicrobial treatment has ceased [28,29]. Enterococci have a highly evolved ability to acquire resistance genes from the same or other bacterial species. This can result in the development of multidrug resistance to many drugs used to treat humans and animals [30,31]. The spread of antimicrobial resistance genes in poultry meat production systems in relation to therapeutic use of antibiotics can lead to various challenges ranging from food safety problems to serious human illness. Therefore, monitoring antimicrobial resistance of enterococci in chicken farms is important [6,32].

The use of antimicrobials is currently addressed within the context of the One Health Concept by the World Health Organization (WHO), representing an integrated approach to balancing and optimizing human, animal, and environmental health [33]. The severity of the situation regarding the development and spread of bacterial resistance to antibiotics is addressed through various legislative measures under national and international regulations (both worldwide and in the EU). These measures reflect the commitment to preserving the effectiveness of existing antimicrobials, primarily by using antimicrobials cautiously to treat bacterial infections. Targeted antimicrobial therapy is encouraged, which involves the precise identification of the causative agent of an infection and determining its susceptibility/resistance to antimicrobial agents considered for the initial treatment. The preference is to preserve the effectiveness of selected antimicrobials for human medicine, with restricted and, in some cases, prohibited use in veterinary medicine. These principles are outlined in Regulation 2019/6 of the European Parliament and Council of the EU on veterinary products, which mandates EU member states to gather data on prescribed antibiotic consumption and resistance [34]. As a result, the European Medicines Agency (EMA) categorizes antimicrobials into four categories: A—avoid, B—restrict, C—caution, and D—prudence [35].

Our work is a pilot work that has a relatively wide scope. It shows the spectrum of enterococci species in one-day-old chickens, their prevalence and susceptibility to antimicrobials, and the occurrence of multiresistant strains. This study also compares the results with the data in the literature and suggests connections with the occurrence and spread of these microorganisms as well as their virulence and resistance genes in the environment and in human and animal populations. Above all, the development of antimicrobial resistance in this livestock population due to the overuse of antimicrobials in the veterinary field is a major problem. It is crucial to realize that the micro-ecosystems of animals, plants, humans, and the environment are tightly connected. This work should be followed by further work with a particular focus on the detection of virulence and antimicrobial resistance genes, and the phylogenetic relationships between strains isolated from poultry and humans. A similar work has not yet been published in the Czech Republic.

## 2. Results

### 2.1. Samples and Isolates

Out of a total of 464 samples, 6 isolates (1.3%) originated from broilers, 37 isolates (8.0%) from breeding males, and 421 isolates (90.7%) from breeding females. The sampled chickens belonged to meat hybrids COBB309, COBB500, and ROSS308, as well as the carrier hybrid Lohmann Brown. See Table 1 for detailed data on breeds from individual farms.

### 2.2. Identification of Isolates

A grand total of 128 enterococci isolates (n = 128; prevalence 27.6%) were isolated, spanning 4 species: *E. faecalis* (n = 108; prevalence 23.3%), *E. faecium* (n = 7; prevalence 1.5%), *E. gallinarum* (n = 10; prevalence 2.2%), and *E. hirae* (n = 3; prevalence 0.6%). The MALDI–TOF MS identification scores for all isolates ranged between 2.000 and 2.577. Predominantly, isolates with a growth intensity of 2 and 3 crosses were identified (n = 41 and 33, respectively), constituting 32% and 25.8% of the isolates, respectively. The most significant numbers of isolates were recovered from organs and the yolk sac (n = 40 and 39, respectively), amounting to 31.3% and 30.5% of the isolates, respectively. Solely, *E. faecalis* was isolated from the bone marrow and brain in six and eight cases (5.5% and 7.4% of the isolated strains, respectively). Detailed data can be found in Table 2.

See Table 3 for a depiction of the probability of chicken contamination based on the results of the bacteriological examination of one-day-old chickens.

### 2.3. Antimicrobial Susceptibility Testing

The outcomes of the susceptibility testing demonstrated the high susceptibility of *E. faecalis*, *E. faecium*, and *E. gallinarum* to penicillin, ampicillin, and amoxicillin with clavulanic acid (ranging from 90% to 100%); the combination of sulphamethoxazole with trimethoprim (ranging from 86.1% to 100%); and florfenicol (ranging from 80% to 100%). Conversely, *E. hirae* displayed 0% susceptibility to enrofloxacin and only 33.3% susceptibility to all other tested antibiotics. Detailed results are shown in Figure 1, Figure 2, Figure 3 and Figure 4. However, it must be taken into account that only *E. faecalis* (n = 108) could be correctly evaluated according to the number of tested isolates. Much fewer isolates of other enterococcal species were obtained from the samples examined (*E. faecium* n = 7, *E. gallinarum* n = 10, and *E. hirae* n = 3).

The MIC distribution of individual antimicrobials, including MIC_50_ and MIC_90_, in *E. faecalis*, *E. faecium*, *E. gallinarum,* and *E. hirae* isolates are shown in Table 4. With the exception of *E. hirae* (n = 3), the MIC_50_ and MIC_90_ were below the breakpoints of the susceptibility value of tested enterococci for penicillin, ampicillin, amoxicillin with clavulanic acid, florfenicol, and sulphamethoxazole potentiated with trimethoprim. On the contrary, both of these values were higher than the breakpoints for tetracycline. For enrofloxacin and erythromycin, MIC_50_ was still below the breakpoint, but MIC_90_ was already above the resistance breakpoint. This indicates the danger of further development and spread of phenotypic resistance of enterococci to these antimicrobials, which is higher when there is a greater difference in the values between MIC_50_ and MIC_90_.

Furthermore, Table 5 offers intriguing data regarding the occurrence of multiresistant isolates of enterococci. A total of 16 isolates (n = 16; 12.5% of tested strains) belonged to 3 species of enterococci that exhibited resistance to 3 or more antimicrobials (*E. faecalis*, *E. hirae,* and *E. gallinarum*). Notably, *E. faecalis* had the highest number of multiresistant isolates (n = 12; 11.1% of isolated *E. faecalis* strains), while no multidrug-resistant (MDR) strains were observed in *E. faecium* (n = 0). Strikingly, four isolates (n = 4; 3.1% of isolated enterococci strains) demonstrated resistance to all eight antimicrobials simultaneously. Simultaneous resistance to three antimicrobials was noted in eight isolates (n = 8; 6.3% of isolated strains), while three isolates (n = 3; 2.3% of isolated strains) exhibited simultaneous resistance to five antimicrobials. One isolate (n = 1; 0.8% of isolated strains) displayed simultaneous resistance to seven antimicrobials.

The occurrence of multidrug-resistant isolates on individual farms is shown in Table 6. MDR isolates were detected at varying degrees on six farms. The most common MDR strains of enterococci were found on 3 farms 1 Bil, 9 Mel, and 8 Mal (n = 6, n = 4, and n = 3, respectively) with a prevalence of 28.6, 25.0, and 10.3%, respectively, in hybrids ROSS 308, and COBB 500. The chickens on these farms did not originate from the same source. However, all isolates that were resistant to all eight tested antimicrobials (n = 4) originated from the same farm (1 Bil) and belonged to the species *E. faecalis* (n = 2) and *E. hirae* (n = 2).

### 2.4. Statistical Analysis

Statistically significant differences in susceptibility were found in the case of penicillin, ampicillin, amoxycillin with clavulanic acid, and florfenicol only between *E. faecalis* and *E. hirae* isolates (all *p* < 0.05). Similarly, statistically significant differences were found when comparing susceptibilities between *E. faecalis* and *E. gallinarum*, between *E. faecium* and *E. hirae,* and between *E. gallinarum* and *E. hirae* (all *p* < 0.05) in the case of enrofloxacin. On the contrary, no statistically significant differences in susceptibility between individual species of enterococci were demonstrated in the case of erythromycin, tetracycline, and sulphamethoxazole with trimethoprim. Differences in the frequency of susceptible and resistant strains within the *E. faecalis* species only were also statistically highly significant (*p* < 0.01). The results of the statistical calculations are shown in the Appendix A.

## 3. Discussion

It is unequivocal that the prevalence of distinct species of enterococci, along with their levels of antimicrobial resistance and multiresistance, tends to correlate with the specific type of material and the kind of animals from which the strains are isolated [36,37,38,39,40]. In our study, the enterococci prevalence of 27.6% in one-day-old chickens is comparable to prior reports of 29.75% and 28.32% in broiler chickens [41] but remains relatively low when compared to the prevalence in wild animals [38]. This observed pattern could be influenced by various factors, including geographical and climatic regions, economic statuses, regional legislations, animal husbandry practices, and the rational utilization of antimicrobials, which manifests in the form of antimicrobial resistance in the captured strains [11,40,42,43,44].

Due to various virulence factors, enterococci can cause nosocomial infections, urinary tract infections (UTIs), wound infections, and bacteraemia in the human population. They are also common causes of endocarditis, pelvic and intra-abdominal infections, otitis, sinusitis, septic arthritis, endophthalmitis, and occasionally throat and brain infections [11]. Among human infections, *E. faecalis* is the most frequently isolated species (85–90% of cases), followed by *E. faecium* (5–10% of cases) [36]. More recent studies indicate a shift in favour of *E. faecium*. For instance, in 2023, a total of 75 enterococcal strains were isolated and the results confirmed the presence of *E. faecalis* in 55% of human disease cases and *E. faecium* in 45% of cases in India [45].The difference between the prevalence of *E. faecalis* and *E. faecium* in our study (93.9% and 6.1%, respectively) remains relatively substantial, in contrast to some literature data [36,45], which suggest a shift in prevalence in favour of *E. faecium*.

Among the less common types of enterococci, *E. gallinarum*, *Enterococcus avium*, *Enterococcus raffinosus*, *E. hirae*, *Enterococcus mundtii*, *Enterococcus casseliflavus*, and *Enterococcus durans* were isolated from human clinical material in a previous study, with prevalence rates of 6.2, 4.1, 2.5, 2.5, 1.7, 1.2, and 0.8%, respectively [37]. The situation is similar in animals, with *E. faecalis* and *E. hirae* being the most frequent species. This is also supported by the work of Slovak scientists in 2021, who isolated 283 enterococcal strains from wild animals. Among these, the most frequently isolated species were *E. faecalis*, *E. hirae*, *E. faecium*, *E. casseliflavus*, *E. durans*, and *E. mundtii*, with prevalence rates of 67.1, 15.9, 6.4, 4.2, 3.5, and 2.8%, respectively [38].

Some enterococci also act as opportunistic pathogens in birds. In Thailand in 2022, different species of enterococci were detected in cloacal swabs and chicken meat in 29.75% and 28.32% of cases, respectively [41]. In avian pathology, the most significant species are *E. faecalis*, *Enterococcus cecorum*, *E. hirae*, and *E. durans*. Their presence in poultry is associated with endocarditis and liver granulomas in turkeys, arthritis in ducks, amyloidosis in laying hens and broilers, ascites in hens, pulmonary arterial hypertension in broilers, and central nervous system lesions in chickens. Chickens are most susceptible to *E. cecorum* infections at 7–14 days of age. Morbidity rates increase within a few weeks of infection. Clinical symptoms manifest as lameness progresses to paralysis. Pathologically, necrosis of the femoral head, tendinitis, arthritis, and osteomyelitis are detected, especially in the area of the thoracic vertebrae, which bulge dorsally and compress the spinal cord. *E. hirae* or *E. durans* can be found in cases of death associated with nervous symptoms, tremors, torticollis, septicaemia, and encephalomalacia in chicks in the first two weeks of age [39]. In contrast, four species of enterococci were detected in our samples. *E. faecalis* (n = 108; prevalence 23.3%), *E. faecium* (n = 7; prevalence 1.5%), *E. gallinarum* (n = 10; prevalence 2.2%), and *E. hirae* (n = 3; prevalence 0.6%) were isolated from these samples, but no strains of *E. cecorum* were found.

For over 20 years, it has been known that enterococci can acquire resistance to antibiotics relatively easily and can transfer resistance genes to other microorganisms [16]. Enterococci resistant to multiple antimicrobials, including vancomycin, are commonly isolated from humans, wastewater, aquatic habitats, agricultural waste, and animal sources, suggesting their potential to enter the human food chain [46,47].

Various scientific studies report different levels of enterococci resistance to antimicrobials. In the case of poultry, resistance to penicillin identified in isolates from chicken meat products in the USA ranged from 14.1% to 87.5% of cases, while resistance to amoxicillin with clavulanic acid ranged from 0% to 37.5% of cases [40]. In other studies, resistance to ampicillin in chickens varied between 14.6% and 54.7% [42,43], erythromycin resistance was detected from 31.1% to 61.0% [42,44], enrofloxacin resistance reached up to 69.4% [48], chloramphenicol resistance ranged from 0% to 42.7% [42,43], tetracycline resistance ranged from 48.8% to 94.7% [42,43], sulphamethoxazole with trimethoprim resistance ranged from 36.6% to 88.0% [42,48], and vancomycin resistance ranged from 0% to 31.1% [42,43].

In our case, relatively low resistance levels to penicillin, ampicillin, and amoxicillin with clavulanic acid were observed for *E. faecalis*, *E. faecium*, and *E. gallinarum*, with a maximum of 10% of tested strains exhibiting resistance. Conversely, various literature sources report resistance rates of up to 87.5% for this group of antimicrobials [40,42,43]. In the case of tetracycline, our resistance rates ranged from 50.0% to 60.2%, while the literature indicates comparable or higher rates ranging from 48.8% to 94.7% [42,43]. In terms of florfenicol, our resistance rates ranged from 0% to 20%, whereas the literature reports comparable rates ranging from 0% to 42.7% for chloramphenicol [42,43]. Similar trends were observed for erythromycin, with resistance rates ranging from 40% to 73.1% in our study, while literature presents comparable rates ranging from 31.1% to 61% [42,44]. Concerning enrofloxacin, our resistance rates varied between 10% and 52.8%, whereas the literature indicates higher rates (up to 69.4%) [48]. We identified notably lower resistance rates for sulphamethoxazole potentiated with trimethoprim, whereas the literature sources report significantly higher rates ranging from 36.6% to 88% [42,48]. Notably, our findings regarding *E. hirae* strains revealed 100% resistance to enrofloxacin and 66.7% resistance to penicillin antibiotics, tetracycline, erythromycin, florfenicol, and sulphamethoxazole with trimethoprim; however, it is important to acknowledge the limited sample size of only three tested strains. The described susceptibility results of the antimicrobial combination of sulphamethoxazole/trimethoprim are questionable due to the inhibition of folate synthesis, which has a synergistic effect on a wide spectrum of bacterial species. Enterococci are unusual in that they can absorb folic acid from the environment, thereby bypassing the effects of sulphamethoxazole/trimethoprim [49]. Therefore, in vitro susceptibility testing of enterococci to sulphamethoxazole/trimethoprim can provide a susceptible result in a folate-free environment, yet this antimicrobial combination is often ineffective in the treatment of severe enterococcal infections [50,51,52,53]. For these reasons, the susceptibility breakpoint for sulphamethoxazole/trimethoprim is set very strictly: ≤0,03/0,6 mg/L [54]. Our research additionally disclosed a notably low occurrence of multidrug-resistant (MDR) strains of enterococci in one-day-old chickens, with only 16 enterococcal strains, constituting 12.5% of the tested isolates. This contrasts with various literature data indicating MDR prevalence ranging from 24.59% to 97.93%. In our study, triresistance (n = 8; 6.3% of isolated strains) was most frequently detected, followed by octaresistance (n = 4; 3.1% of isolated enterococcal strains), pentaresistance (n = 3; 2.3% of isolated strains), and heptaresistance (n = 1; 0.8% of isolated strains). The existing literature reports pentaresistance and tetraresistance prevalence of 35.05% and 32.98%, respectively [55].

In a previous study, Turkish authors highlight the prevalence of multidrug-resistant (MDR) enterococcal strains—those resistant to three or more antimicrobials simultaneously. Their research reports multiresistance in up to 97.93% of isolated strains, with pentaresistance and tetraresistance being the most frequent, occurring in 35.05% and 32.98% of cases, respectively [55]. Other authors have reported lower prevalence of multiresistance in enterococci over the past decade. For instance, in 2016, a prevalence of 83% was reported for MDR enterococci isolated from meat and fermented meat products [56]. In the same year, another study described a prevalence of 78% in strains obtained from food samples [57]. In 2013, a study published a multiresistance prevalence of 59% in strains isolated from food and a prevalence of only 24.59% prevalence in strains isolated from cheeses in 2015 [58,59]. Given this, the efforts of the professional community in collaboration with governmental bodies to significantly reduce antimicrobial usage in both veterinary and human medicine are entirely understandable.

When assessing reports on the monitoring of bacterial susceptibility/resistance to antimicrobials, we must always take into account that differences in the prevalence of antibiotic resistance in different countries depend on their geographical location, the time of isolation of the disease agent, and the use of particular antibiotics for treatment and prevention in a given farm. However, resistances rates are found at varying degrees worldwide [60].

## 4. Materials and Methods

### 4.1. Samples and Isolates

During the period from 1 August 2021 to 31 July 2023 (24 months), a total of 464 mixed samples from 4640 one-day-old chickens were examined as part of routine inspections at 12 poultry farms in the Czech Republic (refer to Table 7).

The chickens were sourced from hatcheries in Germany, Hungary, and the Czech Republic. The sampling was conducted by randomly selecting 10 chickens from various crates within each shipment, from each supplier, and from each group. These individual samples were then combined to create mixed samples. Upon collection, the samples were promptly frozen at −20 °C and transported in a frozen state to the laboratory, where they were processed within 24 h.

After thawing, sterile sampling was performed using Transbak system swabs containing Amies medium with activated carbon (Dispolab s.r.o. Brno, Czech Republic) within a flow box. The sampling procedure involved removing the skin ventrally after disinfecting it with an ether–alcohol-dampened swab (consisting of 1 part ether and 1 part 65% alcohol) and allowing evaporation of the ether–alcohol residue for 1 min. Swabs were taken from the abdominal wall around the navel, the yolk sac, and organs within the body cavity (lungs, heart, and liver). Additionally, bone marrow culture from dissected femurs in 58 cases and brain culture from chickens in 16 cases were randomly performed.

Culturing examinations were conducted using meat-peptone blood agar (MPBA) and Edwards agar (both Lab Media Servis s.r.o., Jaromer, Czech Republic). The plates were incubated aerobically at 37 ± 1 °C for 18–24 h [61,62]. Growth intensity was semi-quantitatively expressed as crosses (+, ++, +++, and ++++). The colony count values for individual evaluation categories are provided in the description below Table 2.

### 4.2. Identification of Isolates

Any suspect strains were isolated and subsequently confirmed using the MALDI—TOF Mass Spectrometry method, employing the Microflex LT System spectrometer (Bruker Daltonics GmbH, Bremen, Germany). This was assessed using the MBT Compass spectrum Library Revision L 2020 (Bruker Daltonics GmbH, Bremen, Germany). Identification scores (ID) ranging from 2.300 to 3.000 were deemed highly probable species identification, values ranging from 2.000 to 2.299 indicated secure genus identification and probable species identification, values ranging from 1.700 to 1.999 indicated probable genus identification, and values ranging from 0.000 to 1.699 were considered unreliable identification.

### 4.3. Antimicrobial Susceptibility Testing

To assess susceptibility, the microdilution method was used to determine the minimum inhibitory concentration (MIC) of individual antimicrobials. The tested antimicrobials included penicillin G, ampicillin, amoxicillin with clavulanic acid, erythromycin, enrofloxacin, florfenicol, tetracycline, and sulphamethoxazole with trimethoprim. In the laboratory of the Veterinary Research Institute Brno (Czech Republic), diagnostic sets for determining MIC were produced according to the Clinical Laboratory Standards Institute (CLSI) CLSI-VET01-A4 methods [63]. See Table 8 for the dilutions of individual antimicrobials and the interpretation criteria categorizing the isolates as susceptible, intermediate, or resistant to the tested antimicrobials, following the CLSI VET01S [64] or European Committee on Antimicrobial Susceptibility Testing (EUCAST) documents [54].

The tests were evaluated after 16–18 h of incubation at 37 ± 1 °C. The quality of the media, MIC plates, and confirmation methods was verified using reference strains of *E. faecalis* (ATCC 29212), *Escherichia coli* (ATCC 25922), and *Staphylococcus aureus* (ATCC 25923) (all from the Czech Collection of Microorganisms, Brno, Czech Republic).

The MIC_50_ and MIC_90_ values were determined from the cumulative results regarding the lowest antimicrobial concentration in mg/L that inhibits the growth of 50% and 90% of isolates [65].

### 4.4. Statistical Analysis

Statistical analysis was performed using Unistat version 6.0.07 (Unistat Ltd. London, UK). The frequency of susceptible and resistant bacterial strains to individual antibiotics in four different *Enterococcus* spp. Was compared using the chi-squared test via the contingency table method. If the criteria for the chi-squared test were not met, Fisher’s exact test was used. Statistical significance was set at *p* < 0.05.

## 5. Conclusions

In conclusion, by enhancing hatching egg hygiene and maintaining consistent hygienic practices in hatcheries, it is possible to enhance the quality, viability, and health of chickens. This would subsequently result in improved health, immune performance, and productivity of new flocks, leading to a reduction in the use of antimicrobial substances in poultry practices. Consequently, the sorting of hatching eggs at farms and hatcheries is essential. Eggs collected from litter or those that are visibly dirty, deformed, uneven in size, or damaged should not be permitted to enter hatcheries. Effective disinfection of their surfaces prior to hatching is vital. After hatching, sorting chicks according to their weight becomes imperative. Underweight chickens, when grouped with similarly sized counterparts, can still be raised successfully. Conversely, poor uniformity can lead to high morbidity and mortality, necessitating frequent antimicrobial interventions, which incur substantial economic losses. In numerous instances, the root causes can be attributed to economic short-sightedness, human irresponsibility, and negligence across various levels of poultry practice. It is known that enterococcal strains from animals can be not only a source of animal and human diseases, but also a source of virulence and resistance genes for the animal and human microecosystems. The above veterinary measures can also contribute to limiting the spread of pathogenic enterococcal strains and their corresponding genes not only among animals but also in human populations and in the environment.

## Figures and Tables

**Figure 1 antibiotics-12-01487-f001:**
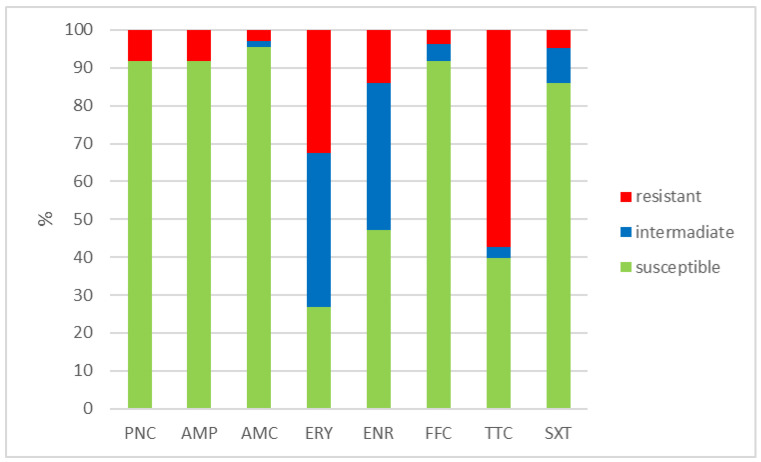
Percentage of susceptible, intermediate, and resistant isolates of *E. faecalis* (n = 108). PNC = penicillin, AMP = ampicillin, AMC = amoxicillin with clavulanic acid, ERY = erythromycin, ENR = enrofloxacin, FFC = florfenicol, TTC = tetracycline, and SXT = sulphamethoxazole + trimethoprim.

**Figure 2 antibiotics-12-01487-f002:**
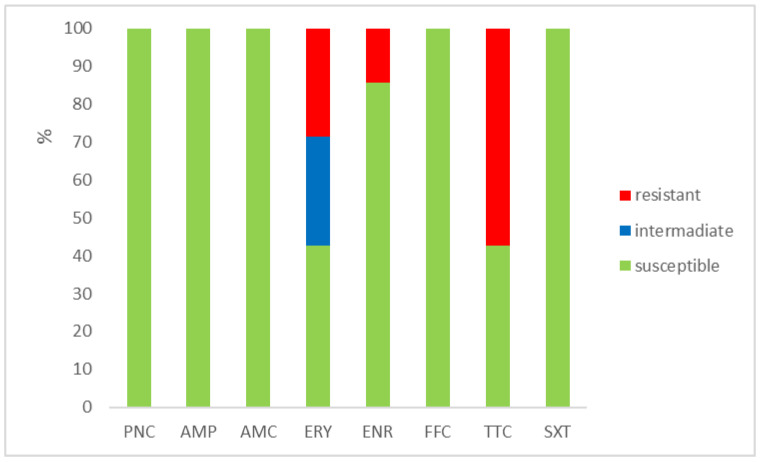
Percentage of susceptible, intermediate, and resistant isolates of *E. faecium* (n = 7). PNC = penicillin, AMP = ampicillin, AMC = amoxicillin with clavulanic acid, ERY = erythromycin, ENR = enrofloxacin, FFC = florfenicol, TTC = tetracycline, and SXT = sulphamethoxazole + trimethoprim.

**Figure 3 antibiotics-12-01487-f003:**
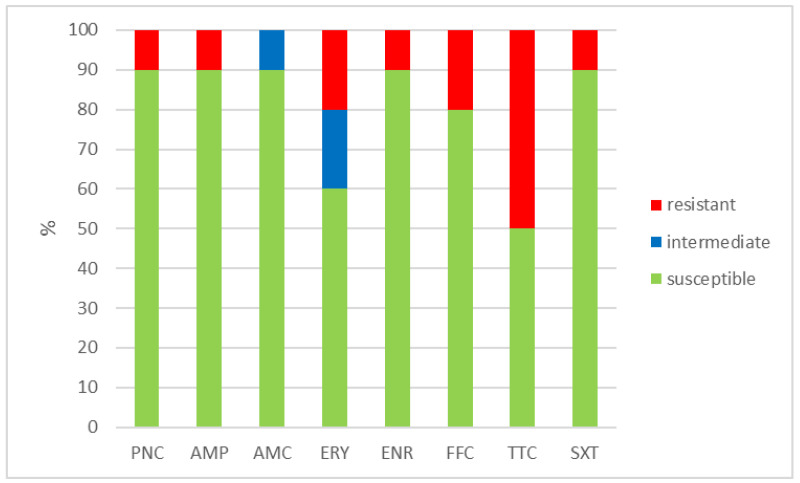
Percentage of susceptible, intermediate, and resistant isolates of *E. gallinarum* (n = 10). PNC = penicillin, AMP = ampicillin, AMC = amoxicillin with clavulanic acid, ERY = erythromycin, ENR = enrofloxacin, FFC = florfenicol, TTC = tetracycline, and SXT = sulphamethoxazole + trimethoprim.

**Figure 4 antibiotics-12-01487-f004:**
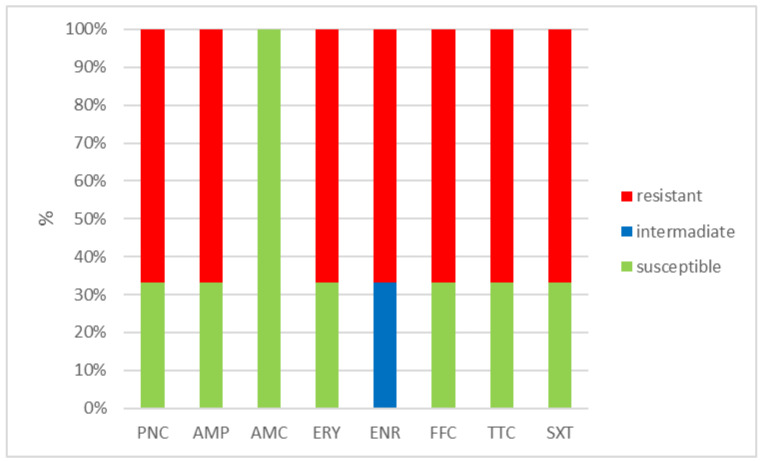
Percentage of susceptible, intermediate, and resistant isolates of *E. hirae* (n = 3). PNC = penicillin, AMP = ampicillin, AMC = amoxicillin with clavulanic acid, ERY = erythromycin, ENR = enrofloxacin, FFC = florfenicol, TTC = tetracycline, and SXT = sulphamethoxazole + trimethoprim.

**Table 1 antibiotics-12-01487-t001:** Breed affiliation and type of production by farms included in the survey.

Farm	Kept Breeds	Reproduction Breeding for Meat	Reproduction Breeding for Eggs	Commercial Laying Hens	Broilers
1 Bil	COBB 500	+	−	−	−
2 Hlu	ROSS 308	+	−	−	−
3 Hor	ROSS 308	−	−	−	+
4 Kun	COBB 500	+	−	−	−
	COBB 309				
5 Kyl	COBB 500	+	−	−	−
6 Lub	ROSS 308	+	−	−	−
7 Maj	ROSS 308	−	−	−	+
8 Mal	COBB 500	−	−	−	+
	ROSS 308				
9 Mel	COBB 500	+	−	−	−
10 Ost	ROSS 308	+	−	−	−
11 Roh	ROSS 308	+	−	−	−
12 Uni	Lohmann Brown	−	−	+	−

+ indicates type of production on individual farm; − indicates that this type of production is not practised on individual farm.

**Table 2 antibiotics-12-01487-t002:** Findings of enterococci and their prevalence (%) in chickens in the 24-month period, and their intensity of growth on plates and in organs, including the percentage of the total number of identified isolates.

Enterococcus Species	*E. faecalis*	*E. faecium*	*E. gallinarum*	*E. hirae*
Parameters				
Number of positive findings (n)	108	7	10	3
Prevalence (%)	(23.3)	(1.5)	(2.2)	(0.6)
Intensity of growth + (%) *	21 (19.4)	4 (57.1)	3 (30.0)	0
Intensity of growth ++ (%) *	33 (30.6)	1 (14.3)	4 (40.0)	3 (100.0)
Intensity of growth +++ (%) *	30 (27.8)	1 (14.3)	2 (20.0)	0
Intensity of growth ++++ (%) *	24 (22.2)	1 (14.3)	1 (10.0)	0
Number of isolates from the navel (%) *	29 (26.9)	2 (28.6)	3 (30.0)	1 (33.3)
Number of isolates from organs (%) *	35 (32.4)	2 (28.6)	2 (20.0)	1 (33.3)
Number of isolates from the yolk sac (%) *	30 (27.8)	3 (42.8)	5 (50.0)	1 (33.3)
Number of isolates from the bone (%) *	6 (5.5)	0	0	0
Number of isolates from the brain (%) *	8 (7.4)	0	0	0

* Number of isolated strains and percentage of the total number of isolates; number of examined strains (n = 464); and intensity of growth, + (1 to 10 colonies per plate), ++ (11 to 50 colonies per plate), +++ (51 to 300 colonies per plate), and ++++ (more than 300 colonies per plate).

**Table 3 antibiotics-12-01487-t003:** Probability of bacterial contamination of one-day-old chickens depending on bacteriological findings.

Origin of Chicken Contamination	Prenatal			Postnatal			
Organ	Ovary	Oviduct	Egg Surface	Skin	Digestive Tract (Drinking)	Respiratory Tract (Aspiration of Aerosol)	Septicaemia or Sepsis
Navel	+	+	+	++	+	−	++
Lung	+	+	+	−	+	++	++
Heart	+	+	+	−	+	++	++
Liver	+	+	+	−	++	+	++
Yolk sac	++	++	++	+	+	−	++
Bone	+	+	+	−	−	−	++
Brain	+	+	+	−	−	−	++

++ High probability; + low probability; and − no probability.

**Table 4 antibiotics-12-01487-t004:** Distribution of MICs, including MIC_50_ and MIC_90_ in *E. faecalis* (n = 108), *E. faecium* (n = 7), *E. gallinarum* (n = 10), and *E. hirae* (n = 3) isolates.

	MIC (mg/L)													MIC_50_	MIC_90_
	0.03	0.06	0.125	0.25	0.5	1	2	4	8	16	32	64	128	(mg/L)	(mg/L)
*E. faecalis*															
PNC			4	1		5	11	74	4	4	5			4	8
AMP					15	16	18	50		1	2	4	2	4	4
AMC					96	3	3	1	2	1	1		1	≤0.5	1
ERY			8	11	10	15	18	11	9	6	20			2	>32
ENR		14	8	29	29	13	3	2		10				0.5	4
FFC					2	10	41	44	2	5	1	3		4	8
TTC					32	6	2		3	4	17	26	15	32	>64
SXT	93	8	2						5					≤0.03	0.06
*E. faecium*															
PNC							1	6						4	4
AMP					2		2	3						2	4
AMC					6	1								≤0.5	≤0.5
ERY			1		2		1			1	2			2	>32
ENR			1	5				1						0.25	4
FFC					1		5	1						2	4
TTC					3					1	1	1	1	16	>64
SXT	7													≤0.03	≤0.03
*E. gallinarum*															
PNC				1		1	2	5			1			4	4
AMP					2	3	1	3		1				1	4
AMC					8	1			1					≤0.5	1
ERY			1	2	3	1	1				2			0.5	1
ENR		1	2	6						1				0.125	0.25
FFC					1		4	3			2			2	>32
TTC					4	1						3	2	1	>64
SXT	9								1					≤0.03	≤0.03
*E. hirae*															
PNC								1		2				>16	>16
AMP							1			1		1		16	>64
AMC					1					1		1		16	>64
ERY					1					1	1			16	>32
ENR				1					2					8	8
FFC							1				1	1		32	64
TTC					1							2		64	64
SXT	1								2					>4	>4

PNC = penicillin; AMP = ampicillin; AMC = amoxicillin/clavulanic acid 2/1; ERY = erythromycin; ENR = enrofloxacin; FFC = florfenicol; TTC = tetracycline; and SXT = trimethoprin/sulphamethoxazole 1/19. Susceptible; Intermediate; and Resistant. MIC = minimal inhibitory concentration. The dilution ranges of individual antimicrobials are delimited in the grey zone. The MIC values in the grey zone indicate MIC values higher than the highest concentration in the range. Values corresponding to the lowest concentration tested indicate MIC values less than or equal to the lowest concentration in the range. The MIC_50_ and MIC_90_ values represent the lowest concentration (mg/L) inhibiting the growth of 50% and 90% of the isolates in the bacterial culture with a density of 10^5^ CFU/mL.

**Table 5 antibiotics-12-01487-t005:** Numbers and percentages of multiresistant isolates of enterococci isolated from one-day-old chickens in a 24-month period.

Profile	Resistant Isolates of *E. faecalis*		Resistant Isolates of *E. faecium*		Resistant Isolates of *E. gallinarum*		Resistant Isolates of *E. hirae*		Resistant Isolates Total	
	No.	%	No.	%	No.	%	No.	%	No.	%
No resistant isolates	7	6.5	1	14.3	4	40.0	0	0	12	9.4
Number of strains with a mix of intermediary and susceptible results on the tests	30	27.8	1	14.3	0	0	1	33.3	32	25.0
ERY	7	6.5	1	14.3	1	10.0	0	0	9	7.0
ENR	1	0.9	1	14.3	0	0	0	0	2	1.6
TTC	30	27.8	1	14.3	2	20.0	0	0	33	25.8
SXT	1	0.9	0	0	0	0	0	0	1	0.8
ERY, ENR	2	1.9	0	0	0	0	0	0	2	1.6
ERY, TTC	9	8.3	2	28.6	0	0	0	0	11	8.6
ENR, SXT	1	0.9	0	0	0	0	0	0	1	0.8
TTC, SXT	4	3.7	0	0	1	10.0	0	0	5	3.9
PNC, AMP, ERY *	3	2.8	0	0	0	0	0	0	3	2.3
ERY, ENR, TTC *	1	0.9	0	0	0	0	0	0	1	0.8
ENR, FFC, TTC *	0	0	0	0	1	10.0	0	0	1	0.8
ENR, TTC, SXT *	3	2.8	0	0	0	0	0	0	3	2.3
PNC, AMP, ERY, FFC, TTC *	2	1.9	0	0	1	10.0	0	0	3	2.3
PNC, AMP, AMC, ERY, ENR, TTC, SXT *	1	0.9	0	0	0	0	0	0	1	0.8
PNC, AMP, AMC, ERY, ENR, FFC, TTC, SXT *	2	1.9	0	0	0	0	2	66.7	4	3.1
Total of tested strains	108	100.0	7	100.0	10	100.0	3	100.0	128	100.0

* Multiresistance = resistance to three or more antimicrobials, PNC = penicillin, AMP = ampicillin, AMC = amoxicillin with clavulanic acid, ERY = erythromycin, ENR = enrofloxacin, FFC = florfenicol, TTC = tetracycline, and SXT= sulphamethoxazole + trimethoprim.

**Table 6 antibiotics-12-01487-t006:** The distribution of enterococci species and MDR strains on individual farms.

Farm Identification	1 Bil	2 Hlu	3 Hor	4 Kun	5 Kyl	6 Lub	7 Maj	8 Mal	9 Mel	10 Ost	11 Roh	12 Uni	Total Isolates
Isolates													
*E. faecalis*	17	9	13	6	6	4	2	27	14	5	1	4	108
*E. faecium*			2							2	3		7
*E. gallinarum*	2	1				1		2	2		2		10
*E. hirae*	2					1							3
Total strains	21	10	15	6	6	6	2	29	16	7	6	4	128
MDR 3 resist.	2	1	1					2	2				8
MDR 5								1	2				3
MDR 7						1							1
MDR 8	4												4
Total MDR strains	6	1	1			1		3	4				16

MDR 3 = multiresistance to 3 antimicrobials, MDR 5 = multiresistance to 5 antimicrobials, MDR 7 = multiresistance to 7 antimicrobials, and MDR 8 = multiresistance to 8 antimicrobials.

**Table 7 antibiotics-12-01487-t007:** Numbers of samples, chickens, and farms examined in the 24-month (from 1 August 2021 to 31 July 2023) period focusing on *Enterococcus* spp.

Year	Number of Examined Mixed Samples	Number of Examined Chickens	Number of Farms	Number of Examined Broilers		Number of Examined Males		Number of Examined Females	
				n	Percentage of All Examined Chickens	n	Percentage of All Examined Chickens	n	Percentage of All Examined Chickens
2021	140	1400	10	0	0	80	5.7	1320	94.3
2022	210	2100	11	50	2.4	60	2.9	1990	94.8
2023	114	1140	6	10	0.88	230	20.2	900	78.9
Total	464	4640	12 *	60	1.3	370	8.0	4210	90.7

* The number of farms is not possible to add up because the farms are repeated year to year and only some farms are added up.

**Table 8 antibiotics-12-01487-t008:** Tested antimicrobials and breakpoints used for *Enterococcus* spp.

Antimicrobials	Tested Concentrations (mg/L)	*Enterococcus* spp.			Source
		MIC Breakpoints (mg/L)			
		≤S	I	≥R	
penicillin	0.125–16	8	-	16	VET01S
ampicillin	0.5–64	4	8	16	EUCAST
amoxicillin/clavulanic acid	0.5/0.25–64/32	4	8	16	EUCAST
erythromycin	0.125–16	0.5	1–4	8	VET01S
enrofloxacin	0.06–8	0.25	0.5–1	2	VET01S
florfenicol	0.5–64	8	16	32	VET01S
tetracycline	0.5–64	4	8	16	VET01S
trimethoprim/sulphamethoxazole	0.03/0.6–4/76	≤0.03/0.6	0.06/0.125–1/19	2/38	EUCAST

S = susceptible; I = intermediate; and R = resistant.

## Data Availability

Raw data supporting the conclusions of this study are available from the authors upon request.

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
