# Peer review of "Enterococci Isolated from One-Day-Old Chickens and Their Phenotypic Susceptibility to Antimicrobials in the Czech Republic"

_antibiotics, 2023, doi:10.3390/antibiotics12101487_

Round 1

Reviewer 1 Report

Comments for authors

The manuscript entitled “Enterococci in Day-Old Chickens and Their Susceptibility to Antimicrobials” described the isolation and antibiotic resistance profiles of Enterococcus spp. in day-old chickens. This MS generates the general report and information on enterococci isolates for One Health Concept. Some flaws need to be revised as follows:

1.       Line 13: Remove “conclusion”

2.       Line 58: italicize “E. faecium

3.       For all names of bacteria, authors can use an abbreviated name for the second, third, fourth, ….. mentions. For example, “Enterococcus faecalis” for the first mention in Line 21. “E. faecalis” for the second mention in Line 24, and “E. faecalis” for the third mention in Line 55. Kindly check all names of bacteria and use abbreviated names in the whole manuscript with the consistency of writing.

4.       Line 59-61: You can use “E.” instead of “Enterococcus

5.       Line 51-111: There are many reports provided in the Introduction section. It would be better if this literature were divided into the Discussion section.

6.       Line 164: italicize “E. hirae

7.       Line 176: remove “species”

8.   The Discussion part should be improved as described.

9.       Line 200: italicize “E. faecium

10.   Line 230: “farms”

11.   The format of tables is ambiguous, this issue should be revised.

-          Heading: Table 2, Table 4, Table 5,

-          Direction of text: Table 3, Table 6

12.   MIC values for each isolate (total of 128 isolates) should be provided as a supplementary file.

Good Luck!!!

Some words and sentences should be improved by native English users.

Author Response

Reviewer 1:

The manuscript entitled “Enterococci in Day-Old Chickens and Their Susceptibility to Antimicrobials” described the isolation and antibiotic resistance profiles of Enterococcus spp. in day-old chickens. This MS generates the general report and information on enterococci isolates for One Health Concept. Some flaws need to be revised as follows:

  1. Line 13: Remove “conclusion”

It was removed.

  1. Line 58: italicize “E. faecium

The correction was performed.

  1. For all names of bacteria, authors can use an abbreviated name for the second, third, fourth, …..mentions. For example, “Enterococcus faecalis” for the first mention in Line 21. “E. faecalis” for the second mention in Line 24, and “E. faecalis” for the third mention in Line 55. Kindly check all names of bacteria and use abbreviated names in the whole manuscript with the consistency of writing.

Yes, correct reminder. The problem was solved.

  1. Line 59-61: You can use “E.” instead of “Enterococcus

Yes, the abbreviation was used.

  1. Line 51-111: There are many reports provided in the Introduction section. It would be better if this literature were divided into the Discussion section.

We agree, the part of the text has been transferred to Discussion.

  1. Line 164: italicize “E. hirae

The italics were used.

  1. Line 176: remove “species”

It was removed.

  1.  The Discussion part should be improved as described.

The chapter Discussion was improved.

  1. Line 200: italicize “E. faecium

The italics was used.

  1. Line 230: “farms”

It was removed.

  1. The format of tables is ambiguous, this issue should be revised.

-          Heading: Table 2, Table 4, Table 5,

-          Direction of text: Table 3, Table 6

All tables were revised and corrected.

  1. MIC values for each isolate (total of 128 isolates) should be provided as a supplementary file.

This table was replaced to the supplementary file.

Good Luck!!!

Reviewer 2 Report

Please see the attached file for the Reviewer's Report on the manuscript ID: antibiotics-2607577.

Lots of English language, expression, and typing errors are in the manuscript, which should be carefully and totally edited and revised throughout the manuscript, necessarily helped by professional English editing service.

Author Response

Reviewer 2:

The authors surveyed the prevalence of Enterococci in day-old chickens collected from a total of 464 mixed samples from August 1, 2021 to July 31, 2023 in the Czech Republic. A total of 128 strains were isolated, including Enterococcus faecalis, Enterococcus faecium, Enterococcus gallinarum, and Enterococcus hirae, with prevalence rates of 23.3%, 1.5%, 2.2%, and 0.6%, respectively. The results showed that high percentages (80-100%) of E. faecalis, E. faecium, and E. gallinarum strains were susceptible to penicillins, sulfamethoxazole with trimethoprim, and florfenicol. Conversely, all E. hirae strains displayed resistance to enrofloxacin, with high degrees of resistance to other tested antimicrobial. Among the isolated strains, 16 showed resistance to three or more antimicrobial, and four to all eight antimicrobial agents tested.

The Materials and Methods, and Results sections of this manuscript are very preliminary, and they are not the sections in the scientific article. Moreover, some issues should be clarified.

Essential extensive revisions of this manuscript should be totally done.

Essential revisions

  1. In the title, the source information of the chicken should be provided.

The title has been clarified.

  1. In the Introduction, please provide the information of previous studies on the prevalence and spectrum of Enterococci spp. in chickens in the Czech Republic, and describe the significance of this study.

No similar study on this topic had previously been published in the Czech Republic. Therefore, it would be very beneficial to publish this work after the necessary modifications.

  1. In the Materials and Methods:

(1) The authors should use subtitles to logically present the materials and methods.

 This problem was dissolved. This chapter has been clearly divided.

(2) The authors stated that “Upon collection, the samples were promptly frozen at -20°C and transported in a frozen state to the laboratory, where they were processed within 24 hours”. Please cite the method used for the sample collection. Did the frozen condition of the samples reduce the recovery of the Enterococcus strains?

Methods of the samples freezing at -20°C for bacteriological examination are described in professional literature, e.g. in the case of milk, and are routinely used in practice today. Some works show that the viability of mainly Gram-positive pathogens is preserved in these samples for up to several weeks. Our experience confirms this even in the case of dead chickens. After all, in many cases, enterococci were isolated from chicken cadavers even in strong intensity (four crosses). In addition, all our samples were frozen under the same conditions before cultivation and therefore the results are comparable.

Some citations of works on effect of samples freezing:

Schukken, Y.H.; Smit, J.A.H.; Grommers, F.J.; Vandegeer, D.; Brand, A. Effect of Freezing on Bacteriologic Culturing of Mastitis Milk Samples. J Dairy Sci 1989 72(7), 1900-1906. DOI: 10.3168/jds.S0022-0302(89)79309-7

Murdough, P.A.; Deitz, K.E.; Pankey, J.W. (1996) Effects of freezing on the viability of nine pathogens from quarters with subclinical mastitis. J Dairy Sci 1996,79(2), 334-336. DOI:10.3168/jds.S0022-0302(96)76368-3

Costa, E.O.; Melville, P.A. ; Ribeiro, A.R.; Pardo, R.B.; White, C.R. Microbiological method for diagnosis of clinical bovine mastitis. Arq Bras Med Vet Zootec 1997, 49(2), 159-167. WOS:A1997XM76100003

 (3) Please cite the method used for the isolation of the Enterococcus species.

Methods of the isolation and the confirmation of enterococci are generally known and are commonly used by veterinary and human laboratories. These methods are also listed in various textbooks and standard operating procedures, for example:

Votava, M.; Ruzicka, F.; Woznicova, V.; Cernohorska, L.; Dvorackova, M., Hola, V.; Zahradnicek O. Medical microbiology - examination methods (In Czech), 1st ed.; Publisher: Neptun Brno, Czech Republic, 2010, pp. 312-313. ISBN:978-80-86850-04-8

Bzdil, J. (Ptacy s.r.o., Valasska Bystrice, Czech Republic) SOP02/21 Cultivation and identification of bacteria from the genus Enterococcus, Standard operating procedure(In Czech), 2021. 

However, we do not know whether the editors will accept references published in the Czech language.

(4) Lines 57-62, and 105-109: please provide more detailed information regarding the samples from which the Enterococcus strains were isolated.

All samples on lines 57-62  were obtained from human clinical materials and samples on lines 105-109 originate from food.

  1. Some essential data are missing in the results. For example,

(1) The distribution of the 128 Enterococci strains in different sampling farms.

Information has been added in the table.

(2) The source of the MDR strains, and the four strains resistant to all eight antimicrobial agents tested.

Information has been added in the same table.

(3) The phylogenetic relationships of the 128 Enterococci strains should be analyzed.

 The genotyping has not yet been performed. Our work is a pilot work on this topic. The following article will include a detailed description of the virulence and antimicrobial resistance genes and a dendrogram will also be created.

(4) Additionally, the authors should logically present the results in subtitles.

The chapter Results was more clearly divided into paragraphs.

  1. All the Tables should be presented according to the guidance of the journal. I suggest the authors to read published articles in the journal Antibiotics, and learn how to prepare Tables.

All tables were corrected accordance to journal guidelines.

  1. Abbreviations and acronyms are typically defined the first time the term is used within the abstract and again in the main text and then used throughout the remainder of the manuscript. Please consider adhering to this convention, and check throughout the manuscript.

Yes, that is a correct reminder. Thanks for it! The problem was dissolved.

  1. Lots of English language, expression, and typing errors are in the manuscript, which should be carefully and totally edited and revised throughout the manuscript, necessarily helped by professional English editing.

Yes, we understand. The final language proofreading will be carried out in cooperation with the editorial staff.

Reviewer 3 Report

Authors conducted the study “Enterococci in Day-Old Chickens and Their Susceptibility to Antimicrobials” appears to have valuable insights into the antimicrobial resistance, and multi-resistance patterns of enterococci in day-old chickens in the Czech Republic. However, there are several shortcomings and areas that could be improved in the manuscript:

Comment 1: Authors should a brief introduction in the abstract section that provides context for why studying enterococci in day-old chickens is important. While the abstract briefly mentions the methods i.e. important information such as the sampling techniques, the number of samples per farm, and the criteria for selecting the 12 poultry farms should be included for clarity and transparency. The abstract lacks the statistically significant differences in susceptibility/resistance between species. Authors should include this information which would help readers assess the robustness of the findings. The abstract could conclude with a sentence or two suggesting potential implications of the findings and directions for future research or interventions.

Comment 2: The introduction covers a wide range of topics related to enterococci, their attributes, roles, virulence factors, and potential impacts on human and animal health. However, the information could be structured more logically to guide the readers through the different aspects of the topic. However, the introduction could benefit from a more explicit statement of the research gap or problem that this study aims to address. What knowledge or understanding is currently lacking in this area?

Comment 3: It is suggested for authors to improve the results for clarity and comprehensiveness with the help of visual aids such as graphs, charts, or diagrams. If suitable, consider incorporating these visual aids to represent trends, patterns, or comparisons more intuitively. The Results section should not be just a collection of data tables and descriptions. It's an opportunity to explain the significance of the findings and relate them back to your research questions and objectives.

Comment 4: In the discussion section of the manuscript, while the authors mentioned that the prevalence of enterococci and their resistance patterns are influenced by various factors, authors should elaborate on each factor's potential impact. For example, how do geographical and climatic regions affect prevalence and resistance rates? How do different animal husbandry practices contribute to variations in antimicrobial resistance? Providing more in-depth explanations will enhance the reader's understanding. It would be beneficial to discuss the potential consequences of the observed resistance patterns in day-old chickens. Authors have mentioned the limited sample size for E. hirae strains. The authors should discuss the limitations of the study, and also discuss how this limitation could impact the generalizability of your findings for this particular species. Since the study deals with antimicrobial resistance, discussing the implications of the findings on antibiotic use in poultry farming is important. Do the results suggest the need for changes in antibiotic usage practices? Are there potential strategies that could be adopted to mitigate resistance?

Comment 5: Authors should ensure the consistency in the abbreviations and units used throughout the study. For example, you use both "μg/mL" and "mg/L" to express concentrations. Stick to a single format for clarity.

Minor English editing is required. Authors should improve it grammatically as well.

Author Response

Reviewer 3:

Authors conducted the study “Enterococci in Day-Old Chickens and Their Susceptibility to Antimicrobials” appears to have valuable insights into the antimicrobial resistance, and multi-resistance patterns of enterococci in day-old chickens in the Czech Republic. However, there are several shortcomings and areas that could be improved in the manuscript:

Comment 1: Authors should a brief introduction in the abstract section that provides context for why studying enterococci in day-old chickens is important. While the abstract briefly mentions the methods i.e. important information such as the sampling techniques, the number of samples per farm, and the criteria for selecting the 12 poultry farms should be included for clarity and transparency. The abstract lacks the statistically significant differences in susceptibility/resistance between species. Authors should include this information which would help readers assess the robustness of the findings. The abstract could conclude with a sentence or two suggesting potential implications of the findings and directions for future research or interventions.

We tried to add some information, however the Abstract chapter is limited to 200 words. That is why we have moved information about statistical methods and other required informations to the Results chapter.

Comment 2: The introduction covers a wide range of topics related to enterococci, their attributes, roles, virulence factors, and potential impacts on human and animal health. However, the information could be structured more logically to guide the readers through the different aspects of the topic. However, the introduction could benefit from a more explicit statement of the research gap or problem that this study aims to address. What knowledge or understanding is currently lacking in this area?

We have tried to add information mainly about the restriction of the use of antimicrobials in animals. This is a very hot topic of the present time.

Comment 3: It is suggested for authors to improve the results for clarity and comprehensiveness with the help of visual aids such as graphs, charts, or diagrams. If suitable, consider incorporating these visual aids to represent trends, patterns, or comparisons more intuitively. The Results section should not be just a collection of data tables and descriptions. It's an opportunity to explain the significance of the findings and relate them back to your research questions and objectives.

We tried to create graphs showing the level of antimicrobial resistance in the captured enterococci species, however, these graphs increased the volume of our work and do not contain accurate percentage data. That's why we kept the existing table of sensitivities and added it to the supplementary files. We have added new connecting and explanatory information to our text.

Comment 4: In the discussion section of the manuscript, while the authors mentioned that the prevalence of enterococci and their resistance patterns are influenced by various factors, authors should elaborate on each factor's potential impact. For example, how do geographical and climatic regions affect prevalence and resistance rates? How do different animal husbandry practices contribute to variations in antimicrobial resistance? Providing more in-depth explanations will enhance the reader's understanding. It would be beneficial to discuss the potential consequences of the observed resistance patterns in day-old chickens. Authors have mentioned the limited sample size for E. hirae strains. The authors should discuss the limitations of the study, and also discuss how this limitation could impact the generalizability of your findings for this particular species. Since the study deals with antimicrobial resistance, discussing the implications of the findings on antibiotic use in poultry farming is important. Do the results suggest the need for changes in antibiotic usage practices? Are there potential strategies that could be adopted to mitigate resistance?

How climate affects the appearance and spread of various living organisms is a very current topic in scientific literature and in the media today. These relationships are rather complicated and are currently the subject of scientific studies. Of course, it is certainly possible to apply them also to microorganisms. Also, the level of health and veterinary care and education certainly has an effect on the morbidity of people and animals and is also reflected in the consumption of antimicrobial substances and the subsequent level of resistance of isolated microorganisms. It is clear that in developing and developed countries, where there is a different level of hygiene, breeding and feeding, the situation is completely different. In developing countries, there is probably not enough legislation regarding ATM treatment, there is a lack of modern technology, the system of prevention, sanitation of breeding facilities and premises does not work properly, which can increase the potential for the spread of resistance and virulence genes. In the Discussion, we tried to comment on other topics that you describe.

Comment 5: Authors should ensure the consistency in the abbreviations and units used throughout the study. For example, you use both "μg/mL" and "mg/L" to express concentrations. Stick to a single format for clarity.

We unified the expression of abbreviations of microbial generic names in the text. First mentions are written in full family names, and subsequent mentions are given in abbreviations. The concentrations of antimicrobials in the text have been unified.

Reviewer 4 Report

It is a well written article.

I will address my concerns as follows:

1. Why did the authors used two sources for antimicrobial resistance breakpoints.

2. Sensitivity to trimethoprim sulfamethoxazole is questionable due to the intrinsic ability of Enterococcus genus to obtain folates from an external source – This should be addressed and discussed.

3. Please change in the title and everywhere in the text to one-day-old chickens to avoid confusion.

4. Why didn’t the authors performed a molecular analysis of antimicrobial resistance genes in the strains they isolated. The lack of molecular data reduces the quality of the paper unfortunately. Please address this issue in the text.

5. Since the authors tackled the issue of bacteriocin perhaps a molecular analysis of the genes responsible for those should have been done too. Otherwise, the relevance of that information is questionable.

6. Add a paragraph detailing the limitations of this study and perhaps some arguments why is this research relevant, what brings new to the table, originality etc.

Author Response

Reviewer 4:

It is a well written article.

Thank you very much! It is a great honor for us.

I will address my concerns as follows:

  1. Why did the authors used two sources for antimicrobial resistance breakpoints.

Some sources do not contain all the necessary reference values, so it was necessary to draw from several sources.

  1. Sensitivity to trimethoprim sulfamethoxazole is questionable due to the intrinsic ability of Enterococcus genus to obtain folates from an external source – This should be addressed and discussed.

Yes, this information was added and used in the chapter Discussion.

  1. Please change in the title and everywhere in the text to one-day-old chickens to avoid confusion.

It was changed.

  1. Why didn’t the authors performed a molecular analysis of antimicrobial resistance genes in the strains they isolated. The lack of molecular data reduces the quality of the paper unfortunately. Please address this issue in the text.

The genotyping has not yet been performed. Our work is a pilot work on this topic. The following article could be include a detailed description of the virulence and antimicrobial resistance genes and a dendrogram will also be created. In addition, we did not have sufficient funding for these analyses. In the next work could also be possible to describe the probable ways of the transfer of these genes to the environment and human microecosystem.

  1. Since the authors tackled the issue of bacteriocin perhaps a molecular analysis of the genes responsible for those should have been done too. Otherwise, the relevance of that information is questionable.

Yes, you are right. Our future work can also verify the existence of genes responsible for the production of bacteriocins in isolated strains.

  1. Add a paragraph detailing the limitations of this study and perhaps some arguments why is this research relevant, what brings new to the table, originality etc.

We've tried to add this information.

Current work should indicate possible sources of enterococci as pathogens for the environment, for other animal species and especially for humans through poultry products (meat, eggs, feathers) and also through poultry industry wastes (wastewater, rendering products, aerosol dust, litter). There is a danger not only for consumers, but also for nurses and other farm staff and their family members.

Round 2

Reviewer 1 Report

The MS was improved according to my suggestions. The current version can be accepted.

Some phrases should be corrected.

Author Response

Thank you for your review of our manuscript. The responses to all reviewers are in the attached file.

Reviewer 2 Report

Reviewer’s Comments and Suggestions for Authors

(Second Round)

 Journal: Antibiotics, MDPI

Manuscript ID: antibiotics-2607577

Type: Article

Title: Enterococci in Day-Old Chickens and Their Susceptibility to Antimicrobials

Authors: Jaroslav Bzdil, Vladimir Sladecek, David Senk, Petr Stolar, Waicova Zuzana, Nela Kollertova, Monika Zouharova , Katarina Matiaskova and Katerina Nedbalcova*

The authors revised the Manuscript ID: antibiotics-2607577 based on most of my comments and suggestions for authors. However, some essential revisions of this manuscript should still be done.

Essential revisions (The authors’ reply in the First-Round revision shown in bold type)

1. Lines 57-62, and 105-109: please provide more detailed information regarding the samples from which the Enterococcus strains were isolated.

 All samples on lines 57-62 were obtained from human clinical materials and samples on lines 105-109 originate from food.

No such information are shown in Lines 57-62, and Lines 105-109 are deleted in the revised manuscript. Please clarify.

2. Some essential data are missing in the results. For example,

(1) The distribution of the 128 Enterococci strains in different sampling farms.

Information has been added in the table.

Which table? Please clarify.

(2) The source of the MDR strains, and the four strains resistant to all eight antimicrobial agents tested.

Information has been added in the same table.

Which table? Please clarify.

(3) The phylogenetic relationships of the 128 Enterococci strains should be analyzed.

The genotyping has not yet been performed. Our work is a pilot work on this topic. The following article will include a detailed description of the virulence and antimicrobial resistance genes and a dendrogram will also be created.

As reported in my First-Round Reviewers Report on the Manuscript ID: antibiotics-2607577, the Results sections of this manuscript is very preliminary too, therefore, I suggest that the phylogenetic analysis of the 128 Enterococci strains should be analyzed and added in the manuscript.

3. Lots of English language, expression, and typing errors are still in the manuscript, which should be carefully and totally edited and revised throughout the manuscript, necessarily helped by professional English editing service.

Lots of English language, expression, and typing errors are in the manuscript, which should be carefully and totally edited and revised throughout the manuscript, necessarily helped by professional English editing service.

Author Response

The authors revised the Manuscript ID: antibiotics-2607577 based on most of my comments and suggestions for authors. However, some essential revisions of this manuscript should still be done.

Essential revisions (The authors’ reply in the First-Round revision shown in bold type)

  1. Lines 57-62, and 105-109: please provide more detailed information regarding the samples from which the Enterococcusstrains were isolated.

 All samples on lines 57-62 were obtained from human clinical materials and samples on lines 105-109 originate from food.

No such information are shown in Lines 57-62, and Lines 105-109 are deleted in the revised manuscript. Please clarify.

Yes, you are right. We are wery sorry! According to the demands of one of the reviewers, we were forced to move this information to the Discussion chapter (lines 350-352, 356-358 and 435-440). We did not add more detailed information about specific biological materials, because they were quite diverse and would unnecessarily increase the volume of the article. According to the reference links, any reader can easily search for them in Web of Science.

  1. Some essential data are missing in the results. For example,

(1) The distribution of the 128 Enterococci strains in different sampling farms.

Information has been added in the table.

Which table? Please clarify.

The Table 6 gives these informations. We are sorry.

(2) The source of the MDR strains, and the four strains resistant to all eight antimicrobial agents tested.

Information has been added in the same table.

Which table? Please clarify.

These informations are also presented in the Table 6. In addition, we have added some clarifying comments to the text regarding the occurrence of MDR strains on individual farms (lines 306 to 312).  

(3) The phylogenetic relationships of the 128 Enterococci strains should be analyzed.

The genotyping has not yet been performed. Our work is a pilot work on this topic. The following article will include a detailed description of the virulence and antimicrobial resistance genes and a dendrogram will also be created.

As reported in my First-Round Reviewer’s Report on the Manuscript ID: antibiotics-2607577, the Results sections of this manuscript is very preliminary too, therefore, I suggest that the phylogenetic analysis of the 128 Enterococci strains should be analyzed and added in the manuscript.

We are very sorry, but phylogenetic analysis can not be performed at the moment. We do not have enough time and funds for these analyses, however we would like to publish a new article that will include this information. In addition, we would like to compare veterinary and human strains of enterococci with the help of genomic methods in the following work.

  1. Lots of English language, expression, and typing errors are still in the manuscript, which should be carefully and totally edited and revised throughout the manuscript, necessarily helped by professional English editing service.

 Yes, thank you for the reminder, we will definitely use the Language editing services recommended by the editors of Antibiotics journal.

Comments on the Quality of English Language

Lots of English language, expression, and typing errors are in the manuscript, which should be carefully and totally edited and revised throughout the manuscript, necessarily helped by professional English editing service.

We will definitely use the Language editing services recommended by the editors of Antibiotics journal.

Reviewer 4 Report

The authors responded to all my requests. I accept in current form.

Author Response

Thank you for your review of our manuscript. The responses to the reviewers are in the attached file.